# Sulfur-Enriched Bone Char as Alternative P Fertilizer: Spectroscopic, Wet Chemical, and Yield Response Evaluation

**Dana Zimmer** [1,2], **Kerstin Panten** [3], **Marcus Frank** [4,5], **Armin Springer** [4] and **Peter Leinweber** [1,5,*]

1   Soil Science, University of Rostock, Justus-von-Liebig-Weg 6, D-18051 Rostock, Germany; dana.zimmer@io-warnemuende.de
2   Institute for Baltic Sea Research, Seestraße 15, 18119 Rostock, Germany
3   Julius Kühn Institute, Institute for Crop and Soil Science, Bundesallee 69, 38116 Braunschweig, Germany; kerstin.panten@julius-kuehn.de
4   Medical Biology and Electron Microscopic Centre, University Medicine Rostock, Strempelstraße 14, 18057 Rostock, Germany; marcus.frank@med.uni-rostock.de (M.F.); armin.springer@med.uni-rostock.de (A.S.)
5   Department Life, Light & Matter—University of Rostock, 18051 Rostock, Germany
*   Correspondence: peter.leinweber@uni-rostock.de; Tel.: +49-381-498-3120; Fax: +49-381-498-3122

**Abstract:** Phosphorus- (P) rich bone char (BC) could be an alternative P fertilizer in sustainable agriculture; however, it has a low P solubility. Therefore, sulfur-enriched BC (BC$^{plus}$) was tested for chemical composition and fertilization effects in a pot experiment. In BC$^{plus}$ sulfur, concentrations increased from <0.1% to 27% and pH decreased from 8.6 to 5.0. These modifications did not change P solubility in water, neutral ammonium citrate, and citric acid. A pot experiment with annual rye grass (*Lolium multiflorum* L.) and treatments without P (P0), BC, BC$^{plus}$ and triple superphosphate (TSP) was set up. The cumulative dry matter yield of the BC treatment was similar to P0, and that of BC$^{plus}$ similar to TSP. The plant P uptake was in the order P0 = BC < BC$^{plus}$ < TSP. Consequently, the apparent nutrient recovery efficiency differed significantly between BC (<3%), BC$^{plus}$ (10% to 15%), and TSP (>18%). The tested equilibrium extractions, regularly used to classify mineral P-fertilizers, failed to predict differences in plant yield and P uptake. Therefore, non-equilibrium extraction methods should be tested in combination with pot experiments. Additionally, particle-plant root scale analyses and long-term experiments are necessary to gain insights into fertilizer-plant interactions.

**Keywords:** pot experiment; phosphorus; biochar; SEM-EDX; fertilizer value

## 1. Introduction

Knowledge about the limitation of global phosphorus (P) resources sparked an increasing interest in the usage of recycling products as P fertilizers [1]. Additionally, concerns about environmental damages, mainly through diffuse losses caused by open agricultural P cycles [2], led to political actions enforcing nutrient recycling in the European Union (EU) [3] and elsewhere. The P flows in Europe and the recovery potential from waste streams were recently discussed by [4,5]. They emphasized the high P input into the EU due to food and feed import and the inefficiencies of the entire food production and consumption chain with high losses from the total P input through landfills, surface water, and agricultural soils. According to [5], the EU-27 imported 2392 Gg P in 2005, half of which accumulated in agricultural soils (924 Gg) and half was lost as waste (1217 Gg). Similarly, ref. [4] estimated that 600 Gg of P are currently not being recovered for agricultural use, which represents 43%

of the 1400 Gg of mineral-based P applied to crops. They concluded that it is imperative to change human diet and improve crop and animal nutrient use efficiency. Furthermore, they highlighted the importance to decrease losses and thereby increase nutrient recovery and reuse. A variety of different processes is available for nutrient recovery from various waste streams (e.g., sewage sludge, manure, slaughterhouse waste) [6–8]. However, for most of P recycling products, fertilizer value and agronomic efficiency are unknown.

Wastes are sometimes contaminated with unwanted substances impacting the efficiency and economics of nutrient recovery processes [7]. Bone char (BC) is free from heavy metals and organic contaminants such as pharmaceuticals and is, in addition to P, rich in calcium (Ca) and magnesium (Mg) [9]. Phosphorus in BC is mainly bound in a structure similar to hydroxylapatite (HA) [9,10] in accordance with their origin from bones [11]. Generally, P in HA is of low solubility and, consequently, it cannot be expected that bones or BC have a high fertilization value. It has been demonstrated that P from BC has a relatively low solubility but different crops (wheat, potato, and onion) have a strong influence [12]. Consequently, further attempts were directed to increase the P solubility of BC by a sulfur (S) enrichment leading to so-called BC$^{plus}$ [13,14]. Such S-loaded slow P fertilizers have been tested in a wide range of soils over the past decades (e.g., [15–20]). All studies have emphasized the microbial sulfur oxidation and subsequent acidic reaction in soil to increase P solubility and plant availability (e.g., [18,21–23]). First results demonstrated the P- and S-speciation in BCs before and after a vegetation period in soil [24] and confirmed a higher P solubility from S-loaded BC$^{plus}$ compared to BC [13,14]. However, the chemical P solubility of fertilizers does not necessarily correlate well with P uptake in agronomic experiments [25] and the various BCs have not yet been tested in vegetation experiments.

Common extracts to evaluate P solubility from mineral P fertilizers are, with increasing P-extractability, water, neutral ammonium citrate (NAC), citric acid (CA), and mineral acids (sulphuric and nitric acid) [26,27]. The water extract represents P, which is directly or in a short-term plant-available and is, therefore, used to assess labile P in soils [28,29] and mineral P fertilizers [30]. The NAC extract is commonly used to estimate plant-available P from mineral P fertilizers (e.g., [31,32]), dissolving mono- and dicalciumphosphates (MCP and DCP), some Al- and Fe-phosphates and some alkaline Ca-phosphates such as HA ([33] and references therein). Because plants excrete low molecular weight organic acids such as citric acid (e.g., [34]) to solve nutrients from soil [35], citric acid (CA) is used as extracting agent to assess P availability to plants in soils and fertilizers [31,36]. CA is assumed to extract MCP, DCP, some Al- and Fe and some silico-phosphates ([33] and references therein). Mineral acids (MA) such as a mixture of nitric and sulfuric acid is assumed to extract nearly total P such as in the *Aqua regia* (AR) extract ([33] and references therein) and is therefore also used as an extractant for total P [37]. Traditionally, these chemical extracts are used to analyze and classify mineral P-fertilizers produced from rock phosphates according to national and international fertilizer recommendations [26,27]. The ability of these analyses to evaluate the P solubility from recycled P-fertilizers such as BCs has yet to be evaluated.

Furthermore, bulk analyses may not provide sufficient information about the distribution of nutrient elements at the microscopic and submicroscopic scale of plant-microbe-soil-interfaces at which the nutrient mobilization, -transport and -uptake take place. Electron microscopy with elemental detection is a powerful tool to get deeper insights into the composition of particles at this scale (e.g., [38–41]). Total elemental composition has been reported from electron microscopic investigations of BCs [24], but not the spatial distributions of elements.

Therefore, the objectives of this study were to investigate the spatial distribution of nutrient elements at single particles of BCs, the chemical extractability of P in bulk samples of BC and BC$^{plus}$ as well as their fertilization potential for grass. We hypothesize that irrespective of spatial nutrient element distribution, BC$^{plus}$ releases more plant-available P than BC and, therefore, grass can take up more P in BC$^{plus}$ than in BC treatments.

## 2. Materials and Methods

### 2.1. Origin of Bone Chars and Experimental Setup

Bone char was purchased from BONECHAR Carvao Ativado do Brasil Ltda., Maringá—PR, Brasil. It has been manufactured by the pyrolysis of de-fatted bovine bones at more than 800 °C. Subsamples of this BC were S-enriched (BC$^{plus}$) by adsorbing reduced gaseous S-compounds (e.g., H$_2$S) from the biogas stream according to the procedure described in patent DE102011010525 [42].

Fertilizing effects of BC and BC$^{plus}$ in comparison to highly water soluble triple superphosphate (TSP) and a zero P treatment (P0) were evaluated in a pot experiment conducted with annual rye grass (*Lolium multiflorum* L., cultivar Bendix). Seeding material was supplied by Rudloff Feldsaaten GmbH (23611 Sereetz, Germany). In each pot, 280 mg of P were added to 6 kg (DM) of an acidic (pH 5.2) sandy silt soil with initially low amounts of available P (24.2 mg kg$^{-1}$ calcium acetate lactate extractable P (soil P$_{CAL}$). That means that 1.962 g of BC, 2.625 g of BC$^{plus}$, and 1.427 g of TSP were thoroughly mixed with the soil before being transferred to the pots (diameter: 20 cm; height: 17.5 cm). Because of the high P content in bone chars, application rates as fertilisers are similar to mineral fertilizers rather than organic fertilizers. About 99% (BC) and 100% (TSP) of the fertilizers had a size above 1.0 mm, but only 92% of the BC$^{plus}$ particles were above 1.0 mm with 7% of the size between 0.5–1.0 mm. Fertilizers were applied in their original form as in agricultural practice.

The experiment was set up in a completely randomized block design with four replicates. To test the effect of the time of fertilizer application on P availability and uptake, five incubation times before seeding were implemented. The incubation was carried out under ambient temperature conditions in a vegetation hall. Fertilization took place in fortnightly steps (18.03., 01.04., 15.04., 29.04., and 13.05.) and was followed by seeding of 30 seeds of annual rye grass at the 13 May. The soil moisture was kept at field capacity between fertilization and seeding as well as during the growth period. The average temperature between the 18 March and 13 May (incubation) was 11.8 °C with the lowest temperature of −2 °C in March and the highest temperature of 31 °C in May. During the vegetation period, an average temperature of 16.6 °C with a minimum of 1 °C and a maximum of 43 °C was recorded. Plants were only exposed to the very high temperatures during short periods of time, whilst the pots were placed in the covered area of the vegetation hall for harvests. No liming took place but all other essential nutrients, including S (as potassium sulphate), were provided as nutrient solutions before seeding and after each cut, except the last one totalling 1900 mg N, 1647 mg K, 650 mg S, 175 mg Mg, 2 mg Zn, 4 mg Mn, 1.6 mg Cu, 1 mg B, 0.2 mg Mo, and 20 mg Fe pot$^{-1}$. In total, seven cuts were carried out between 23 June and 3 November. After the final harvest, plant roots and soil were separated to determine the P content in roots and the remaining available P in the soil.

### 2.2. Wet Chemical Analyses of Fertilizers, Soil- and Plant Samples

To determine the P solubility of fertilizers, these were dried at 50 °C until constant weight and ground to fine powder. Five P extraction methods were carried out with (1) water (P$_{water}$, [27]); (2) neutral ammonium citrate (P$_{NAC}$, [26]); (3) citric acid (P$_{CA}$, 2%; [27]); (4) mineral acid (P$_{MA}$, sulphuric and nitric acid; [26]); and (5) an *Aqua regia* digestion. The *Aqua regia* extract is assumed to be total P and is therefore abbreviated as P$_t$. Colorimetric analysis of extracted P was carried out according to [43] at 882 nm (Specord 50, Analytik Jena, Jena, Germany). Ca (318.1 nm), Mg (279.0 nm) and P (177.4 nm) in the *Aqua regia* extracts were measured by Inductively Coupled Plasma-Optical Emission Spectrometry (ICP-OES, icap 6000, Thermo Fisher, Cambridge, United Kingdom) and are abbreviated as Ca$_t$ and Mg$_t$. The pH values of BC and BC$^{plus}$ were measured in 0.01 M CaCl$_2$ (5 g bone char and 12.5 mL CaCl$_2$), and determined by a pH electrode (pH 540 GLP MultiCal, WTW Wissenschaftlich-Technische Werkstätten GmbH & Co. KG, 82362 Weilheim, Germany). The total C and S concentrations of BC and BC$^{plus}$ were determined by a CNS elemental analyzer (Vario EL Fa. Foss Heraeus, 63450 Hanau, Germany).

Soil samples were air-dried and sieved to a particle size ≤2 mm. Water soluble P (soil P$_{water}$) was extracted according to [44] and plant available P was extracted with calcium acetate lactate (soil P$_{CAL}$)

according to [45]. P concentrations in extracts were measured with ICP-OES (icap 6000, Thermo Fisher, Cambridge, United Kingdom) at a wavelength of 213.6 nm.

Shoots and roots were dried at 60 °C until constant weight was reached and finely ground in a vibration disc mill (Retsch RS1, 42781 Haan, Germany). Phosphorus concentrations were determined after microwave-assisted digestion in nitric acid (CEM MARS, Metthews, NC, USA) with ICP-OES at a wavelength of 177.4 nm.

### 2.3. Electron Microscopic Analyses of Bone Chars

BC and BC$^{plus}$ particles were analyzed using a field emission scanning electron microscope (SEM, MERLIN® VP Compact, Carl Zeiss Microscopy GmbH, 73443 Oberkochen, Germany) equipped with an energy dispersive X-ray (EDX) detector (XFlash6/30, Bruker Nano GmbH, 12489 Berlin, Germany). Analyses of elemental abundance/contents and distribution were done with SEM-EDX Quantax Esprit software (version 1.9 or 2.0, Bruker Nano GmbH, 12489 Berlin, Germany). The char particles were fixed by hot glue on 0.5″ SEM Pin Stubs (agar scientific; Plano GmbH, 35578 Wetzlar, Germany) and coated with carbon under vacuum (EM SCD 500, Leica Microsystems GmbH, 35578 Wetzlar Germany). SEM-images were taken from the selected char particles. One BC particle and one BC$^{plus}$ particle were analyzed (5 keV) for spatial distribution of elements (each in two spots), especially Ca, P and S, at the surface. From the BC$^{plus}$ particle, two areas were mapped for elemental distribution. From this mapping, mean atom percentages of elements were automatically calculated by the SEM-EDX software. All mappings were set to the same counts collected for these EDX analyses to realize a certain comparability of element intensities between samples.

### 2.4. Statistics

The apparent nutrient recovery efficiency (ANR) of the above ground biomass for the different fertilizers used in the pot experiment was calculated by Formula (1) according to [46].

$$ANR\ [\%] = \frac{(P\ uptake\ with\ test\ fertilizer - P\ uptake\ with\ zero\ fertilizer\ )}{P\ applied\ with\ fertilizers} \times 100 \tag{1}$$

The P budget (Formula (2)) for the applied fertilizers was calculated as follows:

$$P\ budget = (available\ soil\ P + fertilizer\ P) - (available\ soil\ P\ at\ harvest + total\ P\ uptake) \tag{2}$$

The total P uptake is the sum of the P uptake by shoots and roots.

To test for significant differences between treatments, analysis of variance (one-way ANOVA) and Tukey-Kramer HSD *t*-test were performed with JMP (Version 12, SAS Institute, Cary, NC, USA). Treatment differences were considered significant at *p* values < 0.05.

## 3. Results

### 3.1. Analyses of Fertilizers

#### 3.1.1. Total Element Concentrations, pH-Values and P Solubility of Fertilizers

The Ca$_t$ concentrations of bone chars (321 and 235 g kg$^{-1}$, in BC and BC$^{plus}$, respectively) were 1.5 to 2 times higher than in TSP (158 g kg$^{-1}$) whereas the P$_t$ concentrations in bone chars were lower, in BC about $\frac{3}{4}$ of that in TSP and in BC$^{plus}$ about the half (Table 1). The molar Ca$_t$/P$_t$ ratios of bone chars (1.7) were more than 2 times higher than that of the TSP (0.6, Table 1). After accumulation of S, the S concentration in BC$^{plus}$ (270 g kg$^{-1}$) was almost 300 times higher than in BC (0.91 g kg$^{-1}$, Table 1). This S accumulation changed the alkaline pH value of BC to an acidic pH of BC$^{plus}$ (Table 1). As was expected, the molar S/P ratio increased from 0.016 in BC to 2.4 in BC$^{plus}$ but the molar Ca/C and P/C ratios were the same in both bone chars.

**Table 1.** Total elemental concentrations; $Ca_t$, $P_t$ and $Mg_t$ from *Aqua regia* extract measured by ICP-OES, in g kg$^{-1}$ and $S_t$, $C_t$ measured by CNS Analyzer. Atom percentage (%) of Ca, P, Mg, S and C in 2 spots measured by SEM-EDX, molar ratios of $Ca_{(t)}/P_{(t)}$, $S_{(t)}/P_{(t)}$, $Ca_t/C_t$, $P_t/C_t$, and pH values in $CaCl_2$ of bone char (BC), S-enriched bone char (BC$^{plus}$) and triple superphosphate (TSP) and proportions of P in water, neutral ammonium citrate (NAC), citric acid (CA) and mineral acid (MA) extract of the BC, BC$^{plus}$ and TSP in percentage (%) of total P concentrations extracted with *Aqua regia*.

| Element | BC | | | BC$^{plus}$ | | | TSP |
|---|---|---|---|---|---|---|---|
| | Wet-Chemical | SEM EDX | | Wet-Chemical | SEM EDX | | Wet-Chemical |
| | | Spot 1 | Spot 2 | | Spot 1 | Spot 2 | |
| $pH_{CaCl2}$ | 8.6 | | | 5.0 | | | n. d. |
| $Ca_{(t)}$ | 321 | 13 | 14 | 235 | 16 | 17 | 158 |
| $P_{(t)}$ | 148 | 6 | 7 | 107 | 7 | 8 | 200 |
| $Mg_{(t)}$ | 6.0 | 1 | 1 | 4.0 | 0.3 | 0.4 | 6.5 |
| $S_{(t)}$ | 0.91 | 0.3 | 0.4 | 270 | 3 | 4 | n. d. |
| $C_{(t)}$ | 104 | 37 | 36 | 82 | 23 | 22 | n. d. |
| molar $Ca_{(t)}/P_{(t)}$ | 1.7 | 2.1 | 2.0 | 1.7 | 2.2 | 2.0 | 0.6 |
| molar $S_{(t)}/P_{(t)}$ | 0.006 | 0.04 | 0.05 | 2.4 | 0.42 | 0.43 | - |
| molar $Ca_t/C_t$ | 0.9 | | | 0.9 | | | - |
| molar $P_t/C_t$ | 0.5 | | | 0.5 | | | - |
| Extract | Percentage (%) of total P concentrations extracted with *Aqua regia* | | | | | | |
| $P_{water}$ | 0.13 | | | 0.74 | | | 87 |
| $P_{NAC}$ | 35 | | | 37 | | | 96 |
| $P_{CA}$ | 72 | | | 79 | | | 101 |
| $P_{MA}$ | 94 | | | 99 | | | 104 |

n. d. = not determined.

The SEM-EDX analyses revealed similar Ca (13% to 17%), P (6% to 8%) and Mg ($\leq$1%) percentages at the surface of both chars (Table 1). Atom percentages of Al, Si, Na and F at the surface of the BC and BC$^{plus}$ were $\leq$1% (not shown). S percentages of BC$^{plus}$ surfaces were around tenfold higher than that of BC surfaces. The molar Ca/P ratios (2.0 to 2.2) according to the SEM-EDX analyses were slightly higher than those of the *Aqua regia* extract (Table 1). Molar Ca/C and P/C ratios of the EDX-spectra were not calculated because chars were coated with carbon to prevent charging during scanning electron microscopy.

The chemical solubility of BC and BC$^{plus}$ was similar (Table 1). The proportions of extracted $P_{water}$ (<1%), $P_{NAC}$ (<40%) and $P_{CA}$ (<80%) in relation to *Aqua regia* soluble P were alike whereas for TSP all extracts solubilised more than 85% (Table 1). The MA extract solubilised nearly the same amount of P as the *Aqua regia* extract (Table 1). Values >100% are explained by analytical errors.

3.1.2. Spatial Distribution of Elements at the Surface of the Bone Chars

The porous structure of BC as well as of the BC$^{plus}$ particles, resulting from the porous structure of the original bones, was clearly visible in the SEM images (Figure 1, images 1.1 to 1.3 and 2.1 to 2.3; see highlighted pore areas in image 2.3). The intensities of P and Ca were similar in BC and BC$^{plus}$ indicated by the dominating blue and green colors with some hot spots of Ca and P in yellow and red color. The darker blue to black areas in the mappings of Ca and P correspond to structures at the surface of the bone char particles, for example small particles (see image 2.1), edges, cracks (see image 2.2) and pores (see image 2.3) being visible in the images in Figure 1 and highlighted by the red circles. The particles discerned at the surface of the chars were rich in Al and Si (maps not shown). The low S-concentration of BC corresponds to the dominating blue to some green colored spots (Figure 1 image 5.1) which contrasted to the more green-yellow up to red colors of BC$^{plus}$ (Figure 1, images 5.2 and 5.3). According to the average spectra of mapping and the spots (Table 1) the S intensities were larger in BC$^{plus}$ by factor about 10 than in BC. Especially in mapping 5.2, the high S accumulation was clearly visible by the dominant green-yellow color with some red hot spots. According to the blue to green color in image 5.1 and the dominant yellow-green color (with only some red hot spots) in image 5.2, the S of BC as well as of BC$^{plus}$ particles seemed to be homogenously distributed at the surface of the chars.

However, in image 5.3, the S seemed to be accumulated heterogeneously in some areas according to the uneven distribution of the blue, green and some red hot spots. Additionally, comparing higher S intensities (yellow to red) in the top of image 5.3 to the lower Ca and P intensities (images 3.3 and 4.3, blue instead of green), the S accumulated from the biogas stream seemed to cover the original Ca and P of the bone char. In image 5.3, the S seemed to be accumulated in the pores of the bone char because the black areas of pores were smaller than in the SEM image 2.3 and in the mappings of Ca (image 3.3) and P (image 4.3).

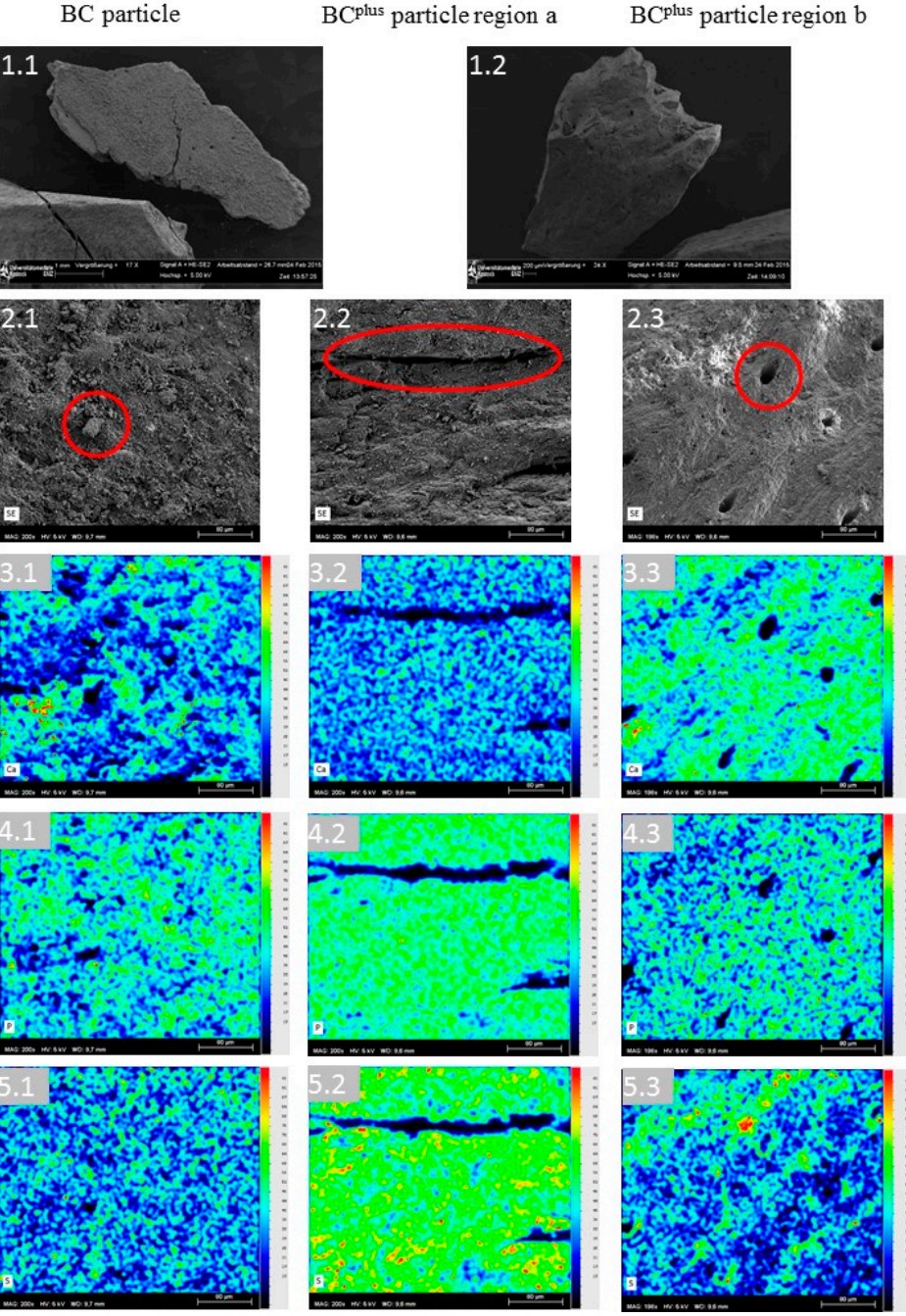

**Figure 1.** Electron microscopic images of BC (image 1.1) and BC$^{plus}$ (image 1.2) and spots (BC: image 2.1, BC$^{plus}$: images 2.2 and 2.3) of mapping and distribution of Ca (BC: 3.1, BC$^{plus}$: 3.2 and 3.3), P (BC: 4.1, BC$^{plus}$: 4.2 and 4.3) and S (BC: 5.1, BC$^{plus}$: 5.2 and 5.3) at the surface of the bone chars BC and BC$^{plus}$ according to the energy dispersive X-ray (EDX) analyses, different colors in the element mappings correspond to different intensities (from low (black) to high (red)) according to the color bar.

### 3.2. Results of the Pot Experiment

#### 3.2.1. Soil pH and Soil $P_{water}$ Concentrations at Seeding and Harvest

At seeding, the soil pH values varied from a minimum of 5.13 to a maximum of 5.33 for all fertilizer treatments across all incubation times, which was comparable to the pH value of 5.20 of the P0 treatment (Figure 2 left). During plant growth, the pH value of the soil decreased by around 0.5 pH units to 4.67 in the P0 treatment. This was slightly lower than the pH values of the fertilizer treatments, which were between 4.76 and 4.79 at harvest (Figure 2). According to the mean pH values at seeding and harvest, the concentrations of $H^+$ (mol $L^{-1}$) increased in the following order: $0.88 \times 10^{-5}$ (BC) $< 1.03 \times 10^{-5}$ (TSP) $< 1.14 \times 10^{-5}$ ($BC^{plus}$) $< 1.5 \times 10^{-5}$ (P0). At harvest, the mean soil pH values of the BC treatment differed significantly from the P0 (4.67) treatment at all incubation times (4.79), except at six weeks (4.73). The $BC^{plus}$ treatment differed significantly at zero weeks (4.80) and two weeks (4.79) and the TSP treatment only at zero weeks (4.81) incubation time from the P0 variant. However, incubation time before seeding did not affect pH values at harvest in the BC, $BC^{plus}$ and the TSP treatments.

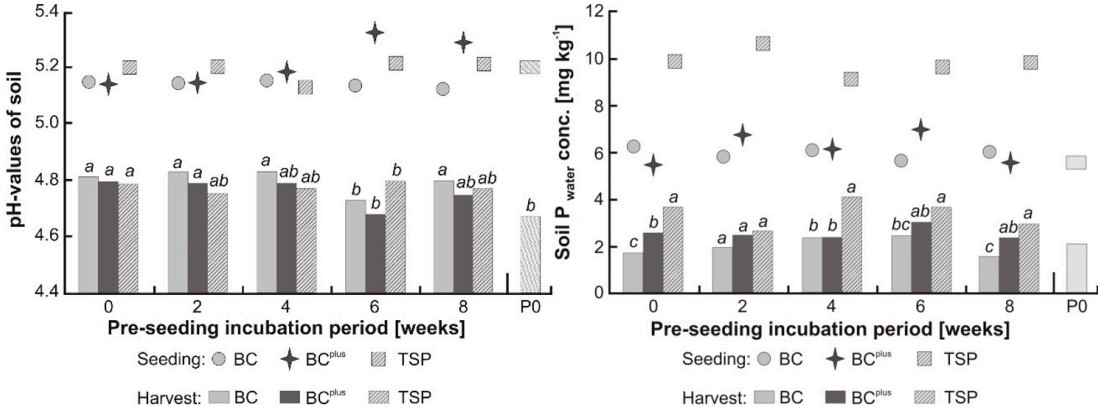

**Figure 2.** Mean pH-values (**left**) and water soluble P (**right**, $P_{water}$) in soil at seeding and harvest at different pre-seeding incubation periods of fertilizers BC, $BC^{plus}$ and TSP in comparison to P0. Differences between fertilizers at harvest, for each pre-seeding incubation period, are indicated by letters and are significant ($p \leq 0.05$) when fertilizers are not connected by the same letter.

The concentration of soil $P_{water}$ was mainly influenced by the type of fertilizer (Figure 2, right). Incubation time had little effect on the soil $P_{water}$ availability of soil at seeding and at harvest. At seeding, the range of soil $P_{water}$ was similar for BC (5.7 to 6.2 mg $kg^{-1}$) and $BC^{plus}$ (5.5 to 7.0 mg $kg^{-1}$) and, therefore, similar to the unfertilized soil (5.5 mg $kg^{-1}$). As expected, TSP significantly increased soil $P_{water}$ to a range of 9.1 to 10.7 mg $kg^{-1}$. At harvest, soil $P_{water}$ significantly differed among fertilizer treatments (Figure 2, right). In all incubation treatments, TSP had most soil $P_{water}$; in four out of five cases, it was significantly higher than in the BC treatment. More soil $P_{water}$ was available in the $BC^{plus}$ than in the BC treatment, except for the four week pre-seeding incubation period. Anyhow, these differences were only significant for the shortest incubation period (Figure 2).

#### 3.2.2. Yield, P Uptake, Apparent Nutrient Recovery Efficiency (ANR) and P Budget

The average dry matter (DM) yield of all pre-incubation periods of the P0 and BC treatments was very similar: 6.0 g $pot^{-1}$ and 6.2 g $pot^{-1}$ at the first cut and 75.5 g $pot^{-1}$ and 74.9 g $pot^{-1}$ cumulative for all seven cuts (Table 2). Contrasting, the yields in the $BC^{plus}$- and TSP treatments were 7.9 g $pot^{-1}$ and 8.0 g $pot^{-1}$ at the first cut and 79.4 g $pot^{-1}$ and 80.4 g $pot^{-1}$ when all cuts were cumulated (Table 2). Even though $BC^{plus}$ and TSP fertilizers gained the highest yields, significant differences were only observed at pre-incubation periods of 4, 6 and 8 weeks (Table 2).

**Table 2.** Analysis of variance (ANOVA) [1] and Tukey-Kramer HSD *t*-test [2] on cumulative aboveground biomass (sum of 7 cuts) in dependence of fertilizer type and incubation time before seeding.

| Fertilizer Treatment | Incubation Time before Seeding [Weeks] | | | | |
|---|---|---|---|---|---|
| | **0** | **2** | **4** | **6** | **8** |
| | Cumulative Biomass Yield [g pot$^{-1}$] | | | | |
| Zero P0 | 76$^a$ | 76$^a$ | 76$^b$ | 76$^{ab}$ | 76$^b$ |
| BC | 77$^a$ | 74$^a$ | 75$^b$ | 73$^b$ | 76$^{ab}$ |
| BC$^{plus}$ | 79$^a$ | 81$^a$ | 76$^b$ | 78$^{ab}$ | 82$^a$ |
| TSP | 78$^a$ | 81$^a$ | 82$^a$ | 80$^a$ | 82$^{ab}$ |
| ANOVA *p*-value | n.s. 0.3190 | * 0.0258 | ** 0.0048 | ** 0.0108 | * 0.0117 |

[1] Significance of p-levels: n.s. = not significant ($p > 0.05$), * = 5% ($p \leq 0.05$), ** = 1% ($p \leq 0.01$), *** = 0.1% ($p \leq 0.001$).
[2] Fertilizers not connected by the same letter are significantly different at the 5% level (Tukey-Kramer HSD *t*-test).

Incubation time of fertilizers before seeding caused no differences in total P uptake by plants for any of the four fertilizer treatments [47]. Nonetheless, the type of fertilizer applied significantly affected the plant P uptake in mg pot$^{-1}$ resulting in the order P0 (155) = BC (156) < BC$^{plus}$ (188) < TSP (214).

Considering that the incubation time had only minor effects on the P uptake by grass, the cumulative P uptake was calculated across all five incubation times to provide a clear indication on the performance of each fertilizer. BC did not increase the P uptake by plants in comparison to the unfertilized control whereas BC$^{plus}$ positively influenced the P uptake even though not as much as TSP (Figure 3).

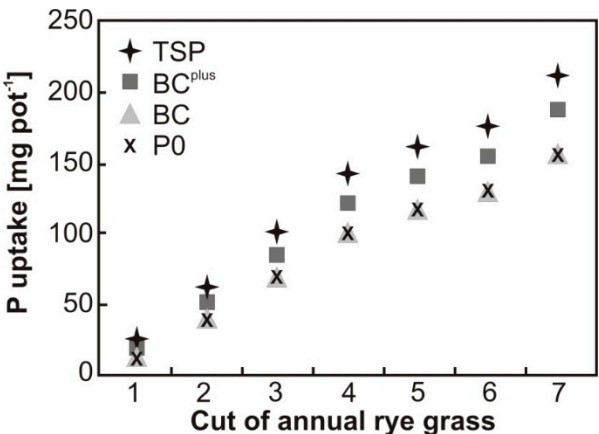

**Figure 3.** Cumulative P uptake of annual rye grass across all incubation times (mean of all incubation treatments from zero to eight weeks).

The average ANR (Formula (1)) was significantly higher for the TSP and lower for the BC treatments following the order BC (<3%) < BC$^{plus}$ (10% to 15%) < TSP (>18%) (Table 3). For BC$^{plus}$ and TSP, the mean ANR was lowest after an incubation of four weeks and highest after eight weeks. For BC, the lowest ANR was after six weeks and the highest also for an incubation time of eight weeks. For BC, the ANR at the incubation times 2 and 6 weeks was negative. However, none of these differences between incubation times were statistically significant.

Besides the P uptake by plants, the soil P$_{CAL}$ and the P budget were also unaffected by the incubation time before seeding. Nonetheless, soil P$_{CAL}$ at seeding was highest for TSP (234 mg pot$^{-1}$), whilst ranging between 139 and 157 mg pot$^{-1}$ for all other treatments (Figure 4), varying considerably between fertilizer treatments. At harvest, soil P$_{CAL}$ concentration followed the order TSP > BC$^{plus}$ > BC = P0. In all fertilization treatments, soil P$_{CAL}$ decreased from seeding to harvest, by 39% for P0, 35% for BC, 27% for BC$^{plus}$ and 29% for TSP (Figure 4). While the P uptake in the total grass biomass

(shoots and roots) of the BC$^{plus}$ treatment was 90% of the TSP treatment, the P uptake was significantly lower in the P0 (73%) and the BC treatments (76%) (Figure 4). Due to the differences in soil P$_{CAL}$ concentrations and P uptake in the total grass biomass, the P budget (Formula (2)) varied significantly between the fertilizer treatments. A negative P budget was observed in the P0 treatment, whereas in the other fertilizer treatments it was increasingly positive, with the highest P surplus recorded for BC (Figure 4).

**Table 3.** Analysis of variance (ANOVA) [1] and Tukey-Kramer HSD *t*-test [2] on apparent nutrient recovery (ANR; %) in dependence of fertilizer type and incubation time before seeding.

| Fertilizer Treatment | Incubation Time before Seeding [Weeks] | | | | |
|---|---|---|---|---|---|
| | 0 | 2 | 4 | 6 | 8 |
| | Apparent Nutrient Recovery [ANR; %] | | | | |
| BC | 1.4$^c$ | −0.3$^c$ | 0.8$^c$ | −1.4$^c$ | 2.2$^c$ |
| BC$^{plus}$ | 10.5$^b$ | 13.4$^b$ | 9.5$^b$ | 11.2$^b$ | 14.7$^b$ |
| TSP | 21.1$^a$ | 21.9$^a$ | 17.9$^a$ | 21.8$^a$ | 23.0$^a$ |
| ANOVA *p*-value | *** 0.0001 | *** <0.0001 | *** 0.0001 | *** <0.0001 | *** <0.0001 |

[1] Significance of p-levels: n.s. = not significant ($p > 0.05$), * = 5% ($p \leq 0.05$), ** = 1% ($p \leq 0.01$), *** = 0.1% ($p \leq 0.001$).
[2] Fertilizers not connected by the same letter are significantly different at the 5% level (Tukey-Kramer HSD *t*-test).

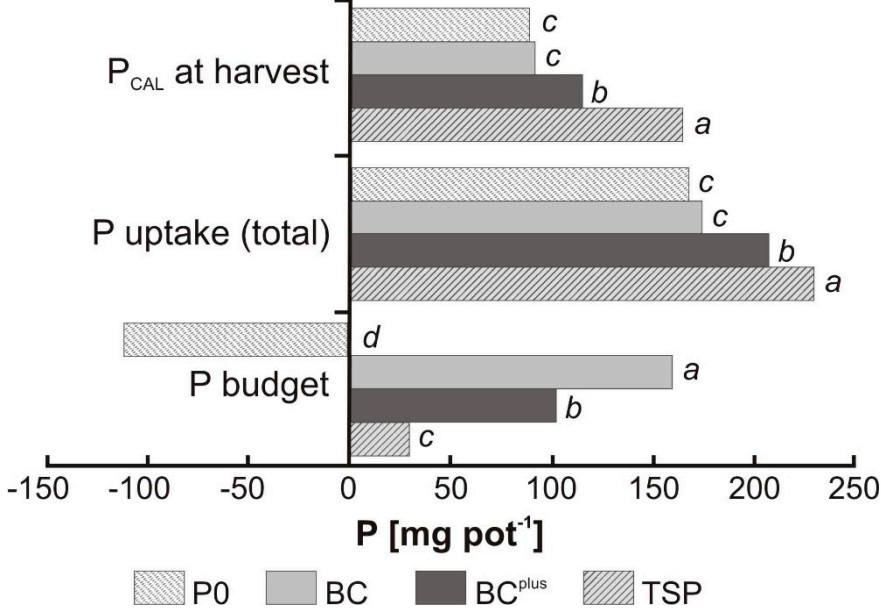

**Figure 4.** Plant available soil P (soil PCAL) at harvest, total P uptake (shoots + roots) and P budget (Formula (2)) of annual rye grass per pot across all incubation times. Levels not connected by the same letter are significantly different at the 5% level (Tukey-Kramer HSD *t*-test).

## 4. Discussion

The BCs of this study had higher Ca$_t$ and P$_t$ concentrations (Table 1) than other biochars from nutrient rich feedstock like e.g., pig manure or sewage sludge (e.g., 24 to 120 g Ca$_t$ kg$^{-1}$ and 16 to 96 g P$_t$ kg$^{-1}$; [48–51]). The P$_t$ and Ca$_t$ concentrations were similar to other bone chars obtained by pyrolysis of rendered materials (60 to 750 °C; P: 86 to 153 g kg$^{-1}$, Ca: 183 to 337 g kg$^{-1}$) as evaluated by [10]. Because P in BCs is mainly bound in a HA-like structure [9,10], this P should be of low solubility and therefore of potentially low plant availability. The S concentration of BC was comparable to that of the bone chars of [52,53], who also detected 0.6 to 1 g S kg$^{-1}$ in chars from meat- and bone-meal. The S in BC resulted from the 0.26% to 0.0.39% natural S in bones as reported by [54,55].

P and Ca were relatively homogeneously spread at the BC and BC$^{plus}$ particle surfaces (Figure 1, images 3.1 to 3.3 and 4.1 to 4.3), caused by their origin from bones, which mainly consist of a nano-crystalline HA lattice dominated by Ca and P [11]. The seemingly lower $P_t$ and $Ca_t$ concentrations in BC$^{plus}$ compared to BC were caused by the accumulation of S at the carbon matrix, because S concentration of BC$^{plus}$ was larger by factor 270 than that of BC (Table 1). Nonetheless, both bone chars had the same $Ca_t/P_t$, $Ca_t/C_t$ and $P_t/C_t$ ratios (Table 1) confirming the same origin and S accumulation at the inner and outer surface of the bone char that "diluted" the other elements.

This accumulation of S in the BC$^{plus}$ was also confirmed by the EDX mapping and atom percentages (Figure 1) because the average spectrum of the BC$^{plus}$ S-mapping had about ten-fold higher S intensities than that of BC. The accumulation of S also occurred in pores (inner surface) of the bone char particles, which was especially visible in the image 5.3 of Figure 1 with reduced black areas of pores in the S-mapping compared to the SEM-image (image 2.3), Ca- (image 3.3), and P- (image 4.3) mappings. The visible heterogeneous distribution of S at the surface of the BC$^{plus}$ (images 5.2 and 5.3 in Figure 1) disagrees with a rather homogeneous S-distribution inside a BC$^{plus}$ particle obtained from a synchrotron-based elemental mapping of cut particles [24]. These differences might be derived from the distribution of BC particles in the biogas stream and the topography of each particle, affecting the contact of the external or internal particle surfaces to the flowing $H_2S$ stream. Thus, it can be highlighted that S accumulation at the BC particles can be heterogeneous at the surface, but S seems to be accumulated not only at the outer surface but also in the pores and therefore at the inner surface of the BC.

The chemical extractions with water, NAC and CA yielded similar P concentrations in BC and BC$^{plus}$ (Table 1) indicating no apparent, short term effect of S-enrichment and acidification on the P solubility. The low to moderate solubility of P in water-, NAC- and CA-extracts of BC$^{plus}$ (Table 1) partly disagreed with [56] who reported a good availability of nutrients and positive plant response of various S-loaded biochars (pyrolysed anaerobically digested solid dairy manure). This disagreement is explained by the different types of source material in the two studies (bones vs. manure).

The P in BC is strongly bound in HA originating from P binding as HA in bones [9,10]. Contrasting, >50% of $P_t$ in cattle manure were easily extractable and identified mainly as inorganic phosphates by $^{31}$P-NMR [57]. They extracted only small amounts of P by HCl, interpreted as less available Ca-bound P. The resulting predominance of rather easily available orthophosphates was confirmed by others, who additionally detected magnesium ammonium phosphate hexahydrate by X-ray diffraction analysis [58,59]. However, pyrolysis reduced the extractability of P compared to that in the source manure, for example, due to the conversion into less-soluble whitlockite [60] and HA [61]. Due to the observed inorganic orthophosphate in manure derived biochar, the author of [62] suggested the predominance of amorphous calcium phosphate. Because the author of [10] detected an increase of HA crystallinity with increasing pyrolysis temperature of bone and our BC was pyrolyzed by 800 °C, it is supposed that differences in P solubility and potential plant availability between bone chars used in this case study and the S-enriched biochars of [56] resulted from differences in the crystallinity of HA.

Soil pH values mutually interact with plant growth and fertilizers and strongly affect nutrient and especially P availability. In this case study, the pH values decreased from seeding to harvest by an average of 0.4 pH units in all fertilizer variants (Figure 2) corresponding to an average increase in $H^+$ concentration (in mol per litre) in the following order: $0.88 \times 10^{-5}$ (BC) < $1.03 \times 10^{-5}$ (TSP) < $1.14 \times 10^{-5}$ (BC$^{plus}$) < $1.5 \times 10^{-5}$ (P0). This general increase in $H^+$ concentrations, which was also detected in the P0 variant, can be explained by plant growth and the release of root exudates [63–66]. The higher pH values of the fertilized pots in comparison to the P0 treatment, observed at harvest, were most likely induced by application of Ca with fertilizers (BC: 630 mg Ca pot$^{-1}$; BC$^{plus}$: 617 mg Ca pot$^{-1}$, TSP: 225 mg Ca pot$^{-1}$) in contrast to the P0 treatment without Ca fertilization. The stronger, but insignificant, mean increase in $H^+$ concentrations in the BC$^{plus}$ treatment compared to the BC and TSP (see above) treatments at harvest were assumed to derive from the further acidification of soil through the S application and microbial oxidation of elemental S to sulfate in soil, which can cause a decrease

in soil pH (e.g., [21,22,67–72]). Such acidic reaction could probably be responsible for dissolving of HA bound-P from BC$^{plus}$ as well as parts of sorbed soil-P. As mentioned above, the decrease in soil pH in the BC$^{plus}$ treatment was not significant. Therefore, it can be speculated that H$^+$ were fast neutralized by dissolution of the HA matrix of the BC$^{plus}$ and/or changes in pH values were restricted to small areas, maybe only some millimetres [73–75] around each BC$^{plus}$ particle. This could imply that bulk soil pH measurement did not reflect small-scale changes around roots or fertilizer particles and that small-scale spatial resolution of pH measurements around BC$^{plus}$ particles would be necessary to verify particle-specific alterations.

Similar to soil pH values, soil P$_{water}$ decreased from seeding to harvest in all treatments (Figure 2, Table 2). Pots fertilized with TSP had the highest soil P$_{water}$ concentration at seeding in comparison to those in the P0, BC and BC$^{plus}$ treatments (Figure 2). However, incubation time did not significantly affect the P$_{water}$ concentration. This disagreed with studies where in calcareous (e.g., [76–78]) as well as in acidic soils [13] the P solubility of highly soluble fertilizers such as TSP and diammonium phosphate decreased with incubation time. This difference could be caused by the fact that fertilizers in the present study were not finely ground but rather used in their original form to simulate field applications. However, significant differences in soil P$_{water}$ were verified between the applied fertilizers at all incubation periods, except in the two weeks (Figure 2). The mostly intermediate soil P$_{water}$ concentrations at harvest for BC$^{plus}$ (Figure 2) demonstrated that the P solubility of BC$^{plus}$ ranged between BC and TSP. Similarly, the author of [72] reported an increase of P solubility along with decreasing pH after application of elemental S in combination with manure. However, such changes in soil P fractions can be restricted to only 2–3 mm around the rhizoplane and no changes can be observed at wider distance from plant roots (e.g., [75]). Therefore, it is speculated that, similar to the above discussed possible small scale changes of the pH values, the P solubility only changed around the BC$^{plus}$ particles by oxidation of elemental sulfur, which was only partially reflected by the bulk soil water extract.

By contrast to the similarity in P solubility by various extractants (Table 1) and the partially small differences in soil pH and soil P$_{water}$, the results of the pot trial showed a much-better performance of the BC$^{plus}$ in comparison to the BC treatment (Tables 2 and 3, Figure 3). The biomass yield of the BC treatment (93% of TSP) confirmed the low P solubility in the fertilizer extracts (Table 1). However, the larger grass dry matter yield of the BC$^{plus}$ treatment (Table 2, Figure 3) and the higher ANR of 10–15% for BC$^{plus}$ (98% of TSP) (Table 3) cannot be explained by the P solubility in the fertilizer extracts (Table 1). Similar discrepancies between chemical P solubility and plant response are well documented in the literature. For example, the authors of [31] described contradictory results for the NAC, CA, and formic acid (FA) extractions and the agronomic effectiveness of different phosphate rock materials. Furthermore, (e.g., [79,80]) reported that the mentioned chemical extracts are often inappropriate to quantify phytoavailable P from biosolids or biochars. These disagreements between results of fertilizer extractions and results of pot experiments (Tables 2 and 3, Figure 3) strongly support that solubility data of common mineral fertilizer extracts do not always reflect the plant availability of P from alternative P fertilizers. This insufficient performance of the chemical extracts is critical because such tests are used according to [26] to assess the P solubility of fertilizers. The discrepancies in our study underline the importance of pot experiments to get insights into soil and plant root-derived effects (e.g., uptake, adsorption, precipitation, (re)dissolution) on plant availability of nutrients (e.g., [81,82]). The lack of common fertilizer extracts to predict real plant availability and uptake of nutrients is probably caused by the equilibrium conditions in the extracts in contrast to the non-equilibrium conditions in soil caused by the nutrient uptake of plant roots. For instance, the diffuse gradients in thin films (DGT) is such a non-equilibrium technique (e.g., [83–85]) and reflected P uptake and biomass yield of maize treated with sewage sludge-based P fertilizers more accurately than standard chemical extraction tests for P fertilizers (e.g., water, citric acid, and neutral ammonium citrate [86]. Finally, it cannot be excluded, that the better performance of BC$^{plus}$ to a certain extent originates from the larger mass of the char applied and general positive effects of soil amendment with reactive surfaces that can store plant nutrients, water or provide microbial habitats.

In an efficient agronomic system, the P budgets should be around zero to avoid negative effects for the environment through fertilization [87]. The negative P budget (Formula (2)) of the P0 treatment of the pot experiment can be attributed to the mobilisation of P from soil (Figure 4), which was also reflected by the reduction of soil $P_{CAL}$ from 145 to 89 mg pot$^{-1}$ from seeding to harvest. Such reductions in soil $P_{CAL}$ were observed for all treatments, confirming the P uptake by plants either from available soil P or P from fertilizers. Positive P budgets indicate that more fertilizer was applied than was taken up by the plants and that the soil still contains plant-available P. This means generally, that the applied P was either not solubilised and therefore not plant-available or that the solubilised P from fertilizer was meanwhile adsorbed by soil compounds (e.g., [76,88]). Confirming the ANR calculations made earlier (Table 3), the P budget obviously was lowest for the highly water-soluble TSP fertilizer, intermediate for the BC$^{plus}$ and highest for the BC treatment (Figure 4). According to this order, it is supposed that P was not adsorbed in sufficient amounts by soil compounds, but P was insufficiently dissolved from the BC fertilizers. However, P dissolution seemed to be higher from the BC$^{plus}$ than from BC according to the P budget and total P uptake (Figure 4). Because of the low P solubility of BC, it cannot be recommended for fertilization of arable crops at this point. Instead, the fertilization value of BC for perennial plants or within a complete crop rotation needs to be evaluated. In contrast, BC$^{plus}$ can have high potential as multi-element P-, Ca- and S-fertilizer, but progression of nutrient release, plant availability, and long-term effects in soil have to be studied in longer-term pots, especially field experiments.

## 5. Conclusions

This study on alternative P fertilizers used a beneficial combination of different analysis methods and a pot experiment to demonstrate discrepancies between common P-extraction techniques for fertilizers and soils and the yield and P uptake by grass. The chemical equilibrium extraction techniques, regularly used to classify mineral P-fertilizers produced from rock phosphates according to national and international fertilizer recommendation regulations, failed to predict differences in plant yield and P uptake between alternative P-fertilizers (BC and BC$^{plus}$) and TSP. Therefore, non-equilibrium extraction methods such as DGT should be tested for alternative P fertilizers in combination with plant growth experiments to evaluate alternative P fertilizers according to EU fertilizer regulations.

Besides bulk soil analyses such as water or CAL extracts, particle-scale analyses are necessary to gain insights into fertilizer particles-plant root-interactions to evaluate local effects. Elemental mapping by SEM-EDX was especially suited to demonstrate the homogeneous distribution of Ca and P according to bone origin and the enrichment of S at the BC$^{plus}$ surface and, therefore, the potential of BC as an S adsorber and S fertilizer. According to the grass yield and ANR of BC$^{plus}$, being similar to TSP, and irrespective of the similar Ca- and P distribution in the two bone chars, BC$^{plus}$ has a potential as P and S fertilizer and should be evaluated also in long-term studies for its nutrient availability and nutrient sorption as well as for further effects in soil amendment. Contrasting, BC actually cannot be recommended as a fertilizer for arable crops because of the lowest yield combined with the highest P budgets, resulting in a potential higher environmental risk for later P leaching from soil, compared to the other tested fertilizers.

**Author Contributions:** Conceptualization: D.Z., K.P., P.L.; Data curation: wet chemical: K.P., D.Z., SEM-EDX: M.F., A.S.; Formal analysis: SEM-EDX mapping: M.F., A.S., D.Z.; Analysis of fertilizers and pot experiment: K.P.; Funding acquisition: P.L., D.Z., K.P.; Investigation: D.Z., K.P., M.F. and A.S.; Methodology: K.P., M.F. and A.S.; Resources: BCs: P.L., pot experiment and wet-chemical analyses: K.P.; Supervision: P.L.; Writing—original draft, D.Z., K.P.; Writing—review & editing, K.P. and P.L., M.F. (SEM-EDX); Project Administration: P.L., D.Z.

**Funding:** This research was funded by the German Federal Ministry of Education and Research (BMBF) in the the BonaRes project InnoSoilPhos (grant number 031A558). Parts of the work of D.Z. were financed by the Leibniz ScienceCampus Phosphorus Research Rostock.

**Acknowledgments:** This research was performed within the scope of the Leibniz ScienceCampus Phosphorus Research Rostock.

**Conflicts of Interest:** The authors declare no conflict of interest. The founding sponsors had no role in the design of the study; in the collection, analyses, or interpretation of data; in the writing of the manuscript, and in the decision to publish the results.

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
