# Peer review of "Sulfur-Enriched Bone Char as Alternative P Fertilizer: Spectroscopic, Wet Chemical, and Yield Response Evaluation"

_agriculture, doi:10.3390/agriculture9010021_

Round 1
Reviewer 1 Report
The paper has proofed bone char could be used as P fertiliser from conquer P shortage. The author used various method to examine the properties of the bone char and the pot test was used to evaluate the bone char fertilisers. The paper has well written. The selected test methods are reasonable. However, there has some minor issues. First, the method part need to place after introduction. Secondly, the reference and context style need to meet with journal standard, for example line 32 “The P flows in Europe and the recovery potential from waste streams were recently discussed by [4] and [5]’. It should be presented like ‘ discussed by Buckwell et al. (2016) and Van DijK et al. (2016). Thirdly, the P concentration of the char could compare with other people’s studies.
Author Response
Reviewer 1 comments and author´s response (in blue)
1. Reviewer
The paper has proofed bone char could be used as P fertiliser from conquer P shortage. The author used various method to examine the properties of the bone char and the pot test was used to evaluate the bone char fertilisers. The paper has well written. The selected test methods are reasonable. However, there has some minor issues.
First, the method part need to place after introduction.
# According to the “Instructions for authors” the following order of chapters is indicated in the point “Manuscript preparation”: Research manuscript sections: Introduction, Results, Discussion, Materials and Methods, Conclusions (optional).
However, after having a look at actually published manuscripts we decided to place the chapter “Material and method” after the chapter “Introduction”.
Secondly, the reference and context style need to meet with journal standard, for example line 32 “The P flows in Europe and the recovery potential from waste streams were recently discussed by [4] and [5]’. It should be presented like ‘discussed by Buckwell et al. (2016) and Van DijK et al. (2016).
# According to the “Instructions for authors” the following presentation of references is indicated in the point “Back matter”:
In the text, reference numbers should be placed in square brackets [ ], and placed before the punctuation; for example [1], [1–3] or [1,3]. For embedded citations in the text with pagination, use both parentheses and brackets to indicate the reference number and page numbers; for example [5] (p. 10). or [6] (pp. 101–105).
Thirdly, the P concentration of the char could compare with other people’s studies.
# The P concentration of the chars is compared with biochars from pig manure or sewage sludge as well as with other bone chars (chapter 3 Discussion, first paragraph).
Reviewer 2 Report
This manuscript presents the use of a sulfur-enriched bone biochar to increase the efficacy of bone biochars as P fertilizer. The manuscript is well written, the results are well presented and discussed. My main concern is related to the experimental design since the doses were normalized to the amount of P in the fertilizer. This is ok, but the amounts of biochar may have been different (at least the dose of BC plus was 1/3 higher than BC) and it would be interesting to show this in the methodology and it need to be mentioned in the discussion that some of the effects may be related to the impact of higher amounts of biochar in soil in BC plus. Was the dose of biochar in a typical application range for organic amendments? If not, this should be mentioned in the text.
Another concern (minor) is the structure of the manuscript. Material and method section needs to be presented before the results (according to journal style)
Minor comments:
Line 14. …to 27%. Do you mean by 27%?
Line 97. Please define REM-EDX
Line 109. Please format the data presented in Table 1
Line 193. Figure 3. The symbol for BC can to be easily identified. Please consider changing it.
Line 328-329. This sentence is quite strong. Please rewrite the sentence to clarify if the meaning of the sentence was related to its use as P fertilizer. Why is it not recommended?
Line 338. Please provide some general description, at least those details that may be relevant to the study. Was S added as H2S stream?
Line 342. Please include the amount of biochar added for each treatment. Was it mixed with the soil? What depth?
Author Response
Reviewer 2
This manuscript presents the use of a sulfur-enriched bone biochar to increase the efficacy of bone biochars as P fertilizer. The manuscript is well written, the results are well presented and discussed.
My main concern is related to the experimental design since the doses were normalized to the amount of P in the fertilizer. This is ok, but the amounts of biochar may have been different (at least the dose of BC plus was 1/3 higher than BC) and it would be interesting to show this in the methodology and it need to be mentioned in the discussion that some of the effects may be related to the impact of higher amounts of biochar in soil in BC plus.
# The reviewer is right. According to the high mass of S-accumulation the P-concentration of the BC decreased relatively when “transformed” to BCplus. For testing effects of P fertilization, it was necessary that all pots received the same amounts of P. Therefore, higher masses of BCplus compared to BC had to be placed into the pots. Additionally, we would like to point out that the amounts of applied bone char were generally small (1.962 g BC and 2.625 g BCplus in 6 kg of DM soil or 0.033 and 0.044 % w/w, respectively) in comparison to other biochars produced from feedstock containing low nutrient amounts like e.g. straw, wood or rice husks mainly used as soil improvers. Therefore, we do not expect general ‘char effects’ after a single application. Nevertheless, we considered this advice of the reviewer and mentioned the possibility of other, more general, biochar effects (e.g. lns 426-429).
Was the dose of biochar in a typical application range for organic amendments? If not, this should be mentioned in the text.
# A sentence was included in the chapter “Materials and methods”: “Because of the high P content in bone chars, application rates as fertilisers are similar to mineral fertilizers rather than to organic fertilizers.”
Another concern (minor) is the structure of the manuscript. Material and method section needs to be presented before the results (according to journal style)
# According to the “Instructions for authors” the following order of chapters is indicated in the point “Manuscript preparation”: Research manuscript sections: Introduction, Results, Discussion, Materials and Methods, Conclusions (optional).
However, after having a look at actually published manuscripts we decided to place the chapter “Material and method” after the chapter “Introduction”.
Minor comments:
Line 14. …to 27%. Do you mean by 27%?
# No, “increase to” is correct because the S-concentration increased from 0.91 g kg-1 (0.091 %) to 270 g kg-1 (27 %)
Line 97. Please define REM-EDX
# Thank you, this was a mistake, we replaced REM-EDX by SEM-EDX. This is explained in the chapter “Material and methods”
Line 109. Please format the data presented in Table 1
# Originally, the table was formatted correctly. Something changed during submission maybe due to different word versions. We will send the correctly formatted table to the journal again.
Line 193. Figure 3. The symbol for BC can to be easily identified. Please consider changing it.
# We have changed the symbols x for PO and grey triangle for BC so that it is obvious that the values were almost identical.
Line 328-329. This sentence is quite strong. Please rewrite the sentence to clarify if the meaning of the sentence was related to its use as P fertilizer. Why is it not recommended?
# We rewrote the sentence and explained the necessity of studying the effect in a whole crop rotation.
Line 338. Please provide some general description, at least those details that may be relevant to the study. Was S added as H2S stream?
# The whole procedure is under a patent (DE102011010525), therefore we hardly can present more details. Moreover, the objective of the study was not the insertion of S to BC but the usage of BCplus as P fertilizer. Although the addition of S by H2S is mentioned in the discussion (paragraph about SEM-EDX mapping), we inserted “(e.g. H2S)” in a sentence in the chapter “Material and methods” for clarifying.
Line 342. Please include the amount of biochar added for each treatment.
# The amounts of applicated bone char and TSP were added.
Was it mixed with the soil? What depth?
# The kind of application into the soil was added.
Reviewer 3 Report
The manuscript entitled “Sulfur-enriched bone char as alternative P fertilizer: spectroscopic, wet chemical, and yield response evaluation” by Zimmer et al. deals with an interesting topic due to the limited reserves of phosphate rock (a non-renewable resource) as fertilizer and the need of alternatives in line with circular economy. The data presented in the manuscript are from pot experiments in which rye grass is fertilised with bone char (2 different types) or triple superphosphate to test bone char fertilization potential for grass. I have some mayor concerns regarding statistical analysis, lack of key information (fertiliser doses?) and density of some sections (please, see the specific suggestions and commentaries as follow).
Besides that, I also have a question about the objective of the use of bone chars. I understand that bone chars are used to increase P solubility and bioavailability of soil P, however, the title mentions bone char as an alternative P fertilizer. If this is an alternative to P fertilizers, should we add P and bone chars/ should we reduce P fertilization if we add bone chars?
Abstract
1-Lines 14-16 “In BCplus sulfur concentrations increased to 27 % and pH decreased to 5.0”. Could the author indicate the variation of the pH? I mean, the pH decreased to 5.0 but the reader does not know the initial value.
2-Lines 19-20 “Consequently, the apparent nutrient recovery 19 efficiency of BC (<4 %) was low but that of BCplus (10…15 %) was similar to TSP (>18 %).” I think the author should improve this sentence mentioning when the differences were significant.
Introduction
1-In general, the Introduction section is concise and clear. I miss some information about different processes (temperature of pyrolysis, enrichment of bones with S) and sources (bones) and their effect on P availability/solubility.
2-Lines 39-41: difficult to follow. I suggest splitting it into two different sentences.
Lines 46-48: a reference is needed here. Why does the author use capital letters for calcium and magnesium?
3-Line 74: use “]” instead of “)” after reference 27.
4-In my opinion the hypothesis is vague. According to Lines 57-58, it is known that BCplus will release more P than BC. Could the author consider including some information regarding spatial distribution of nutrient elements at single particles of BC´s?
Results
1-In this manuscript, the Tukey-Kramer t-test is used instead LSD or Tukey’s HSD test. I would like to know the reason.
2-Lines 97-103: “Ca (13…17 %), P (6…8 %)”. (1) Could the author add the corresponding subscript to each element? (2) I suggest using another symbol (for example “─”) instead of “…” to mention the range of variation obtained with REM EDX.
3-Table 1: How could the author explain that some of the methods extracted more P than the 100 %?
4-Figure 2: This figure should be improved. Standard errors could be included to show the variation in the data presented here and to understand better the significant differences (different letters; this comment is also for Figure 3). I suggest using bars for seeding and harvest and avoid the use of symbols; therefore, Figure 2 should have 4 sub-figures and not 2 as know. The reader could benefit from these changes.
4.1-Figure 2 (but this is a general comment): Were the ANOVA and the post-hoc test done including all the incubation periods or for each one individually? What does the author think about a Split-plot ANOVA including (the treatment) and the incubation time as a factor?
4.2-The size of the X and especially Y axes (i.e. Soil P water) should be higher.
5-Table 2. This table could include the standard error with the different mean values at the incubation times. Stars could be removed from the table (the reader could decide the significance level once the p value is shown; no need to repeat information). The author should consider not using the subscript for the letters according to the Tukey-Kramer-t test to make it easier for the reader to see the differences. Finally, in my opinion this information could be shown as a Figure (mean ± standard errors).
6-Lines 184-201. The author should order these 3 paragraphs (information from Table 2 and Figure 3 is mixed).
7-What about S uptake?
Discussion
1-In my opinion this section is too dense. The author could split some paragraphs into 2 or 3 different paragraphs and consider removing some information to avoid repetition (not necessary to include the values that were detailed in the previous section or the figure/table where they can be found).
2-The second paragraph (lines 246-252) is too vague and imprecise. Bones and manure are different material, of course.
3-Lines 253-255: they belong to material and methods.
-Line 311: delete “technique” (written twice).
Conclusions
1-The first sentence is general. As well-known, there are many methods to analyse and predict P bioavailability, P uptake and crop yield.
Material and methods
1-I think the different subsections included here should follow a similar order as in the Results section: first talk about the properties of the fertilizers, then spatial distribution of elements at the surface of the bone chars, and finally the pot experiment.
2-Line 340: What does “planted” mean? Were the seeds pre-germinated and then transferred to the pots? If the response is affirmative, could the author describe this process here?
3-Line 343: What does “Pcal” mean? I think this is explained in Line 385 for the first time (page 12 of 13 of the manuscript). Please, check for the rest of abbreviations in the whole manuscript (for example Cat, Pt, etc, lines 89-91).
4-Lines 342-343: “In each pot 280 mg of P were added to 6 kg (DM) of an acidic (pH 5.2) sandy silt soil with 342 initially low amounts of available P.” Add the source of P (compound, fertilizer) and where this fertilizer was added (Top soil?). A description of the pots (height/ diameter) is recommended.
5-Line 343: “initially low amounts of available P (24.2 mg kg-1 soil PCAL”. Add a “)” at the end of the sentence.
6-Lines 345-346: “Fertilizers were applied in their original form as applied in agricultural practice.” The amount/dose of each fertilizer should be included (BC, BCplus and TSP).
7-Lines 347-350: “To test the effect of the time of fertilizer application on P availability and uptake, five incubation times before seeding were implemented. The incubation was carried out under ambient temperature conditions. Fertilization took place in fortnightly steps (18.3., 01.04., 15.04., 29.04., and 349 13.05.)”. Could the author define and include ambient temperature conditions during the length of the experiment?
8-Lines 352-356: “all other essential nutrients, including S (in total 650 mg S pot-1 as potassium sulphate), were provided in sufficient amounts before seeding and after each cut except the last one.” The author should indicate the amount and how they provided all other essential nutrients (solution, granules). I think “sufficient amounts” is not a scientific way to express the quantity of nutrients added.
9-Lines 389-390: Could the author add more information about the digestion of shoots and roots and the method used to determine this P content?
10-I miss more information about nutrient bioavailability in the soil before adding any fertilizer.
Author Response
Reviewer 3 comments and author´s response (in blue)
Reviewer 3:
The manuscript entitled “Sulfur-enriched bone char as alternative P fertilizer: spectroscopic, wet chemical, and yield response evaluation” by Zimmer et al. deals with an interesting topic due to the limited reserves of phosphate rock (a non-renewable resource) as fertilizer and the need of alternatives in line with circular economy. The data presented in the manuscript are from pot experiments in which rye grass is fertilised with bone char (2 different types) or triple superphosphate to test bone char fertilization potential for grass. I have some mayor concerns regarding statistical analysis, lack of key information (fertiliser doses?) and density of some sections (please, see the specific suggestions and commentaries as follow).
Besides that, I also have a question about the objective of the use of bone chars. I understand that bone chars are used to increase P solubility and bioavailability of soil P, however, the title mentions bone char as an alternative P fertilizer.
# Here the reviewer is not right. We consider both tested bone chars as potential (alternative) P fertilizers but not as means to increase the soil P solubility and bioavailability. As mentioned in the “Introduction” (line 45 and following) bone char is rich in P and free of contaminants in contrast to often Cd- and/ or U polluted mineral P fertilizers or other waste products such as sewage sludge. Therefore, bone char could be a clean P fertilizer. It is not used to increase P solubility/bioavailability in soil. Possible increase in stable soil organic matter and higher water and nutrient holding capacity of the soil may be positive side effects. Because the HA-bound P in BC is of relative low solubility the S was “added” (BCplus) to provoke an acidic reaction in soil/at the BC particle surface and, thus, increase the P solubility from bone char (line 52 and following).
If this is an alternative to P fertilizers, should we add P and bone chars/ should we reduce P fertilization if we add bone chars?
# P concentration in bone char is high due to the origin from bones (Tab. 1). So, the idea is to add the P-rich (and clean) bone char and reduce all other P fertilizations accordingly because they often originate from non-sustainable sources.
Abstract
1-Lines 14-16 “In BCplus sulfur concentrations increased to 27 % and pH decreased to 5.0”. Could the author indicate the variation of the pH? I mean, the pH decreased to 5.0 but the reader does not know the initial value.
# We added the initial pH-value for comparison: “In BCplus sulfur concentrations increased to 27 % and pH decreased from 8.6 to 5.0.”
2-Lines 19-20 “Consequently, the apparent nutrient recovery efficiency of BC (<4 %) was low but that of BCplus (10…15 %) was similar to TSP (>18 %).” I think the author should improve this sentence mentioning when the differences were significant.
# We rewrote the sentence: “Consequently, the apparent nutrient recovery efficiency differed significantly between BC (<3 %), BCplus (10…15 %), and TSP (>18 %).”
Introduction
1-In general, the Introduction section is concise and clear. I miss some information about different processes (temperature of pyrolysis, enrichment of bones with S) and sources (bones) and their effect on P availability/solubility.
# This relevant information is given in the Materials and methods section and partly in the discussion. As we did not do a technical study of pyrolysis processes, technical details do not need to be presented in the Introduction.
2-Lines 39-41: difficult to follow. I suggest splitting it into two different sentences.
# The sentence was split: “They concluded that it is urgently necessary to change human dietary and to improve crop and animal nutrient use efficiency. Additionally, they highlighted the importance to decrease losses and thereby increase nutrient recovery and reuse.”
Lines 46-48: a reference is needed here. Why does the author use capital letters for calcium and magnesium?
# References was added; thanks, this was corrected.
3-Line 74: use “]” instead of “)” after reference 27.
# Thanks, this was corrected.
4-In my opinion the hypothesis is vague. According to Lines 57-58, it is known that BCplus will release more P than BC. Could the author consider including some information regarding spatial distribution of nutrient elements at single particles of BC´s?
# We accept the reviewer´s statement and therefore reformulated the hypothesis, indicating that spatial distribution perhaps is not very important for the plant P uptake. Some information on the spatial distribution of nutrient elements at single particles is presented in the results section.
Results
1-In this manuscript, the Tukey-Kramer t-test is used instead LSD or Tukey’s HSD test. I would like to know the reason.
# To be precise we used the Tukey-Kramer HSD t-test and made changes in the text accordingly. We consider the LSD test as too liberal because it does not correct the type I error.
2-Lines 97-103: “Ca (13…17 %), P (6…8 %)”. (1) Could the author add the corresponding subscript to each element? (2) I suggest using another symbol (for example “─”) instead of “…” to mention the range of variation obtained with REM EDX.
# This is a SEM-EDX analysis, which means a surface information of relative abundance of the elements measured but not a concentration corresponding to a specific volume or mass (g element per kg material). Therefore, we cannot insert a “t” for the relative abundance of an element. We also corrected this in table 1 by adding brackets for the “t” due to both meanings (relative abundance and total element concentrations).
All „…“ for spans were changed to „to“
3-Table 1: How could the author explain that some of the methods extracted more P than the 100 %?
# Values > 100% can be explained by analytical errors. We added this to the text.
4-Figure 2: This figure should be improved. Standard errors could be included to show the variation in the data presented here and to understand better the significant differences (different letters; this comment is also for Figure 3). I suggest using bars for seeding and harvest and avoid the use of symbols; therefore, Figure 2 should have 4 sub-figures and not 2 as know. The reader could benefit from these changes.
# We disagree with this view and prefer to leave the figure as submitted.
4.1-Figure 2 (but this is a general comment): Were the ANOVA and the post-hoc test done including all the incubation periods or for each one individually? What does the author think about a Split-plot ANOVA including (the treatment) and the incubation time as a factor?
#The ANOVA and post-hoc tests were done individually for each incubation period. So far, we did not find any indication that the incubation time had a substantial influence.
4.2-The size of the X and especially Y axes (i.e. Soil P water) should be higher.
# We enlarged the font of “water”.
5-Table 2. This table could include the standard error with the different mean values at the incubation times. Stars could be removed from the table (the reader could decide the significance level once the p value is shown; no need to repeat information). The author should consider not using the subscript for the letters according to the Tukey-Kramer-t test to make it easier for the reader to see the differences. Finally, in my opinion this information could be shown as a Figure (mean ± standard errors).
# We think the readability of the table is best as it is. Also, we prefer to show these data in a table and not in a figure because data can be compared directly more easily from tables than from figures. Alternative data presentations do not affect the general outcomes of the study.
6-Lines 184-201. The author should order these 3 paragraphs (information from Table 2 and Figure 3 is mixed).
# We agree with the reviewer and ordered the paragraph by splitting the table 2 into two tables. The order is now: Table 2, Figure 3, new Table 3.
7-What about S uptake?
# Indeed, we determined the S uptake as well. However, the manuscript already seems quite complex to us, and our research project is on phosphorus. Therefore, we (1) applied compensation S fertilization (650 mg S per pot as mentioned in the Materials section), and (2) decided not to make the story more complicated by involving S in plants.
Discussion
1-In my opinion this section is too dense. The author could split some paragraphs into 2 or 3 different paragraphs and consider removing some information to avoid repetition (not necessary to include the values that were detailed in the previous section or the figure/table where they can be found).
# We split the discussion (lines 220 to 246) for easier reading in three paragraphs. Values in the (now) first paragraph are from references and cannot be deleted. In the second paragraph we deleted some values that are shown in tables. In other parts referring to the EDX mappings, we gave the numbers of plates in the Fig. 1.
2-The second paragraph (lines 246-252) is too vague and imprecise. Bones and manure are different material, of course.
# Some explanations and references (P compounds in manure and biochar manure) were inserted as a new paragraph.
3-Lines 253-255: they belong to material and methods.
# We changed this.
-Line 311: delete “technique” (written twice).
# Thanks; “technique” was deleted
Conclusions
1-The first sentence is general. As well-known, there are many methods to analyse and predict P bioavailability, P uptake and crop yield.
# Yes, there are many analyses techniques but we wanted to highlight especially the beneficial combination of these complementary techniques instead of selecting one of them. For clarifying this point we rewrote the sentence.
Material and methods
1-I think the different subsections included here should follow a similar order as in the Results section: first talk about the properties of the fertilizers, then spatial distribution of elements at the surface of the bone chars, and finally the pot experiment.
# We partly agree with the reviewer and shifted the chapter of SEM-EDX analysis after the chapter about wet chemical analysis. However, it is important to start with the origin of the material (bone char, soil and plant samples) and following explaining the analytical techniques. Therefore, the chapter “Origin of bone chars and experimental setup” still is the first chapter in “Material and methods.
2-Line 340: What does “planted” mean? Were the seeds pre-germinated and then transferred to the pots? If the response is affirmative, could the author describe this process here?
# We rephrased this sentence. As described in line 352, the grass was seeded.
3-Line 343: What does “Pcal” mean? I think this is explained in Line 385 for the first time (page 12 of 13 of the manuscript). Please, check for the rest of abbreviations in the whole manuscript (for example Cat, Pt, etc, lines 89-91).
# Thanks, we checked abbreviations and explained them, when appearing the first time.
4-Lines 342-343: “In each pot 280 mg of P were added to 6 kg (DM) of an acidic (pH 5.2) sandy silt soil with 342 initially low amounts of available P.” Add the source of P (compound, fertilizer) and where this fertilizer was added (Top soil?). A description of the pots (height/ diameter) is recommended.
# The source of fertilizer is explained in the first sentence of the paragraph: “Fertilizing effects of BC and BCplus in comparison to highly water soluble TSP and a zero P treatment (P0) were evaluated in a pot experiment…”. The application of fertilizers is now explained in the text.
5-Line 343: “initially low amounts of available P (24.2 mg kg-1 soil PCAL”. Add a “)” at the end of the sentence.
# “)” was added.
6-Lines 345-346: “Fertilizers were applied in their original form as applied in agricultural practice.” The amount/dose of each fertilizer should be included (BC, BCplus and TSP).
# The amounts of fertilizers are added in the text.
7-Lines 347-350: “To test the effect of the time of fertilizer application on P availability and uptake, five incubation times before seeding were implemented. The incubation was carried out under ambient temperature conditions. Fertilization took place in fortnightly steps (18.3., 01.04., 15.04., 29.04., and 349 13.05.)”. Could the author define and include ambient temperature conditions during the length of the experiment?
# This is explained in the text.
8-Lines 352-356: “all other essential nutrients, including S (in total 650 mg S pot-1 as potassium sulphate), were provided in sufficient amounts before seeding and after each cut except the last one.” The author should indicate the amount and how they provided all other essential nutrients (solution, granules). I think “sufficient amounts” is not a scientific way to express the quantity of nutrients added.
# This is now explained in more detail in the text.
9-Lines 389-390: Could the author add more information about the digestion of shoots and roots and the method used to determine this P content?
# In the chapter “Material and methods” the information of digestion of plant material and determination of P concentration by ICP-OES was provided:
“Shoots and roots were dried at 60 °C until constant weight was reached and finely ground in a vibration disc mill (Retsch RS1, 42781 Haan, Germany). Phosphorus concentrations were determined after microwave-assisted digestion in nitric acid (CEM MARS, Metthews, USA) with ICP-OES at a wavelength of 177.4 nm.”
10-I miss more information about nutrient bioavailability in the soil before adding any fertilizer.
# In the chapter “Material and methods” the information of potentially plant available P was provided:
“In each pot 280 mg of P were added to 6 kg (DM) of an acidic (pH 5.2) sandy silt soil with initially low amounts of available P (24.2 mg kg-1 soil PCAL).”
Round 2
Reviewer 3 Report
The authors have considered the suggestions made in the first round and included them in the new verison of the manuscript.